# FROM PDES TO WINGBEATS: A NOVEL CONVOLUTIONAL FOURIER LAYER-BASED RESNET MODEL

## ABSTRACT

Recent advancements in Deep Learning apply Fourier Neural Operators (FNOs) for generating numerical solutions of Partial Differential Equations (PDEs). They are efficient due to their global spectral representations. However, their abilities in applied classification or regression tasks for time series have not been studied previously. We further investigate the motivation behind FNOs and provide a more detailed Discrete Fourier Transform-based definition. Furthermore, we introduce *CF-ResNet-1D*, a novel ResNet-inspired model built from Convolutional Fourier Layers being parallel units of FNO and 1D-Convolution. *CF-ResNet-1D* can perform time-series data analysis on raw time-domain signals while also taking advantage of the parallel spectral processing of the FNOs. This combined processing method outperforms spectrogram-based models for insect wingbeat sound classification, achieving state-of-the-art accuracy on benchmark datasets. The outcomes of our research offer promising insights about FNO application in real-world problems, such as mosquito management and the mitigation of insect-related diseases.

## 1 INTRODUCTION

A Convolutional Neural Network (CNN) can learn time series features well using one-dimensional local convolutional kernels (Goodfellow et al., 2016). On the other hand, time series can also be thought of as continuous functions. Therefore, they can be represented in Fourier space and global convolutions can be used to obtain global features.

The Fourier Neural Operator (FNO) (Li et al., 2021) is a modern technique that builds on top of the Discrete Fourier Transform. This neural operator performs learnable transformations in the Fourier domain as a layer-level unit of a deep neural network. These transformations are analogous to convolutions (Kabri et al., 2023) in nature, with the advantage of directly accessible frequency components, thus they are suitable for frequency-based machine learning processes. These operators are extensively used in modeling partial differential equations (PDEs) that govern mathematical and physical simulations providing state-of-the-art engineering solutions (Chaohao et al., 2022; Li et al., 2022; Zhang et al., 2022).

However, only a few studies focus on applying FNOs for classical data science-motivated tasks. Among these according to our best knowledge, only image classification capabilities have been explored (Johnny et al., 2022; Kabri et al., 2023). Both of these studies use FNOs as a global alternative to the convolution operator. While this is a good strategy for processing high-level features, locality is also important in such CNNs. To explore the time series processing-related capabilities of the FNO operator we apply them in the domain of insect wingbeat sound classification.

Every year, around 700 million people are infected and more than one million die from mosquito-borne diseases (Caraballo & King, 2014). Such diseases include malaria, dengue, Zika virus fever, yellow fever, West Nile fever, and encephalitis viruses (Palmer et al., 2017). For each disease, one or more mosquito species are responsible for the transmission. Monitoring is an essential part of control and successful intervention. This is why it is important to classify mosquitoes efficiently and quickly.

Two types of machine learning approaches are applied for this task: audio and image-based. For image-based solutions, classification is based on images of mosquitoes (Okayasu et al., 2019; Motta

et al., 2020; Kittichai et al., 2021). It has previously been shown that the wingbeats of different species have different audio characteristics (Offenhauser & Kahn, 1949), this also justifies the examination of audio or audio-like signals. For audio-based methods, either the raw wingbeat recording is used (Yin et al., 2021), or data obtained after applying some kind of transformation on the raw signals, such as the short-time Discrete Fourier Transform or the Discrete Fourier Transform (Arthur et al., 2014; Chen et al., 2014b; Ouyang et al., 2015; Vasconcelos et al., 2019; Luna-Gonzalez et al., 2020; Wei et al., 2022).

A system was presented to record the wingbeat of insects based on the large aperture optical sensors that turn the light fluctuations into sound (Potamitis & Rigakis, 2016; Rigakis et al., 2019). Here, the light fluctuations are caused by the partial occlusion of light from the wings. Without aiming to be exhaustive, the acquired datasets include namely *Wingbeats* and *Fruitflies*. The *Wingbeats* dataset was investigated using different state-of-the-art deep learning architectures (Wei et al., 2022; Fanioudakis et al., 2018). Fanioudakis et al. (2018) converted the audio signals into spectrograms, and a DenseNet-121 based model achieved 96% test accuracy. However, these results could not be reproduced by Wei et al. (2022). Chen et al. (2014a) have used a very similar technique to capture insect flying sounds, the collected dataset is called *Insects*. Mukundarajan et al. (2017) have shown that even low-cost mobile phones are capable of acquiring acoustic data on mosquito wingbeat sounds. The resulting *Abuzz* dataset was investigated with the same technique and with the same *WbNet* architecture (Wei et al., 2022) as the *Wingbeats* dataset, too.

In our contribution, we provide further details on the mathematical motivation of the FNO for discrete time series processing. One advantage of this task type is that the heuristic justification of the FNO definition is much clearer. We provide a refined definition of the Discrete Fourier Neural Operator that is directly applicable to Deep Learning. We propose a novel Convolutional Fourier Layer by extending FNO's point-wise operation to a proper 1D-Convolution. This Convolutional Fourier (CF) Layer is then used to build ResNet-style models (CF-ResNet-9-1D) that are capable of sound classification. To the best of our knowledge, this is the first attempt that FNO has been used to investigate a real-world time series dataset. We test the proposed model on insect wingbeat sound classification benchmarks, due to the impact of their possible applications.

We run numerical experiments on the *Wingbeats*, *Fruitflies*, *Insects*, and the *Abuzz* datasets for evaluation using the above-mentioned state-of-the-art spectrum-based MobileNet, DenseNet-121, and *WbNet* architectures. We also report the performance of classical 1D ResNet architectures and vanilla FNO-based ResNet architectures trained on raw time-series data. We repeat experiments by Wei et al. (2022) and Fanioudakis et al. (2018) as their original implementation does not differentiate between [1] validation and test data, thus it contains an inherent data leakage which results in misleading test performance.

Our proposed model achieves state-of-the-art results with the largest one overperforming all of the baselines on the majority of benchmark datasets. Furthermore, Convolutional Fourier Layers significantly improve the performance compared to the original FNO implementation by Li et al. (2021).

The paper is structured as follows. In Section 2, we provide some background materials concerning the definition of the Discrete Fourier Neural Operator and the investigated *Wingbeats*, *Fruitflies*, *Insects* and *Abuzz* datasets. Also, the proposed *CF-ResNet-9-1D* architecture is described in detail, then the corresponding classification results are presented in Section 3. Finally, we summarize our findings in Section 4.

## 2 METHODS

### 2.1 DISCRETE FOURIER NEURAL OPERATOR

In this subsection, we discuss the motivation behind defining a crucial component of our proposed architecture: the Fourier Neural Operator, which is based on the work by Li et al. (2021). In this motivation part, rigorous mathematical precision is ignored. Although, the precise definition of the operator is given at the end of this subsection.

---

[1]Reevaluated experiment solving data leak in the original implementation.

Let the numbers $I, O \in \mathbb{N}$ and the functions $r \in L^1\left(\mathbb{R}; \mathbb{R}^{O \times I}\right)$, $u \in L^1\left(\mathbb{R}; \mathbb{R}^I\right)$ be given. Here, $r$ is called as the kernel function. The convolution of these functions $v = r * u \in L^1\left(R; \mathbb{R}^O\right)$ is obtained by the next formula.

$$v(x) = (r * u)(x) = \int_{\mathbb{R}} r(x - y)\, u(y)\, \mathrm{d}y, \quad x \in \mathbb{R}. \tag{1}$$

Applying the convolution theorem the following equation holds.

$$v(x) = \mathcal{F}^{-1}(\mathcal{F}(r) \cdot \mathcal{F}(u)) = F_r(u)(x), \quad x \in \mathbb{R}. \tag{2}$$

The operator $F_r$ in equation (2) is called the Fourier Neural Operator. Assume that $D = (-\pi, \pi)$ and $r \in L^2\left(D; \mathbb{R}^{O \times I}\right)$, $u \in L^2\left(D; \mathbb{R}^I\right)$ also hold true. Here, we assume the quadratic integrability holds per component. Let $u(x) = (u_1(x), u_2(x), \dots, u_I(x))^T$, or shortly $u(x) = (u_j(x))^T$ given by its Fourier series componentwise, i.e. $u_j(x) = \sum_{k \in \mathbb{Z}} U_{k,j} e^{ikx}$. Similarly, we suppose that $r(x) = (r_{l,j}(x)) \in \mathbb{R}^{O \times I}$, where $r_{l,j}(x) = \sum_{k \in \mathbb{Z}} R_{k,l,j} e^{ikx}$. The following equations are used.

$$\mathcal{F}(x \mapsto e^{ikx})(y) = \sqrt{2\pi}\delta(y - k) \quad \text{and} \quad \mathcal{F}^{-1}(y \mapsto \sqrt{2\pi}\delta(y - k))(x) = e^{ikx}, \tag{3}$$

where $\delta$ is the Dirac-delta distribution. Let us assume that the Fourier transformation can be carried out term by term in the Fourier series of the $r$ and $u$ functions. In this case, $\mathcal{F}(r)(y) \in \mathbb{C}^{O \times I}$ and $\mathcal{F}(u)(y) \in \mathbb{C}^I$. With this assumption, we investigate their product, namely $\mathcal{F}(r)(y) \cdot \mathcal{F}(u)(y) \in \mathbb{C}^O$. The $l$-th component of this vector is to be calculated as follows.

$$[\mathcal{F}(r)(y) \cdot \mathcal{F}(u)(y)]_l = \sum_{j=1}^{I} \mathcal{F}(r_{l,j})(y) \cdot \mathcal{F}(u_j)(y) \tag{4}$$

It is necessary to calculate $\mathcal{F}(r_{l,j})(y)$ and $\mathcal{F}(u_j)(y)$, which, taking the Fourier transform elementwise in the Fourier series satisfy the following equations.

$$\mathcal{F}(r_{l,j})(y) = \mathcal{F}\left(\sum_{k \in \mathbb{Z}} R_{k,l,j} \cdot e^{ikx}\right)(y) = \sqrt{2\pi} \sum_{k \in \mathbb{Z}} R_{k,l,j}\delta(k - y), \tag{5}$$

similarly

$$\mathcal{F}(u_j)(y) = \mathcal{F}\left(\sum_{k \in \mathbb{Z}} U_{k,j} \cdot e^{ikx}\right)(y) = \sqrt{2\pi} \sum_{k \in \mathbb{Z}} U_{k,j}\delta(k - y). \tag{6}$$

Using the identities in the formulas (5)-(6), equation (4) can be further modified as follows.

$$\begin{aligned}
&[\mathcal{F}(r)(y) \cdot \mathcal{F}(u)(y)]_l \\
&= \sum_{j=1}^{I} \left(\sqrt{2\pi} \sum_{k \in \mathbb{Z}} R_{k,l,j}\delta(y - k)\right) \cdot \left(\sqrt{2\pi} \sum_{k \in \mathbb{Z}} U_{k,j}\delta(y - k)\right) \\
&= 2\pi \sum_{j=1}^{I} \sum_{k \in \mathbb{Z}} R_{k,l,j} \cdot U_{k,j}\delta(y - k) = 2\pi \sum_{k \in \mathbb{Z}} \sum_{j=1}^{I} R_{k,l,j} \cdot U_{k,j}\delta(y - k).
\end{aligned} \tag{7}$$

At the last equation, it was assumed that the two sums can be exchanged. Let us note that this is the Fourier transform of the function $v(x) = (v_1(x), v_2(x), \dots, v_O(x))$, $v \in L^2\left(D; \mathbb{R}^O\right)$, given by the following Fourier series, assuming that we can take its Fourier transform term by term in its Fourier series.

$$v_l(x) = \sum_{k \in \mathbb{Z}} \sum_{j=1}^{I} R_{k,l,j} \cdot U_{k,j}\sqrt{2\pi}e^{ikx}. \tag{8}$$

Now, we take finite-dimensional parameterizations of both $u$ and $r$ by truncating their Fourier representation at the maximum number of modes $|k| \le k_{\max}$. Thus, we write $\mathbf{R} = (R_{k,l,j}) \in \mathbb{C}^{k_{\max} \times O \times I}$ directly as a complex-valued tensor and similarly $\boldsymbol{U} = (U_{k,j}) \in \mathbb{C}^{k_{\max} \times I}$. We may assume this, because both the function $u$ and the kernel function $r$ are real-valued, and we want $v$ to be real-valued as well. Therefore, we impose conjugate symmetry on the coefficients of the Fourier series of $r$ and $u$, i.e. $R_{k,l,j} = \overline{R_{-k,l,j}}$ and $U_{k,j} = \overline{U_{-k,j}}$ where $k \in \mathbb{Z}$, $|k| \le k_{\max}$ and $j = 1, \dots, I$, $l = 1, \dots, O$. After this heuristic approach, we introduce the following definition.

**Definition 2.1** *Assume that $k_{max} \leq [\frac{n}{2}] + 1$ and let the Discrete Fourier Neural Operator $\hat{F}_{\mathbf{R}, \, k_{max}}$ : $\mathbb{R}^{n \times I} \rightarrow \mathbb{R}^{n \times O}$ defined by the equation*

$$\hat{F}_{\mathbf{R}, \, k_{max}}(\boldsymbol{U}) = \hat{\mathcal{F}}^{-1}\left(\mathbf{R} \cdot \hat{\mathcal{F}}(\boldsymbol{U})\right) \in \mathbb{R}^{n \times O},$$

*where $\mathbf{R} \in \mathbb{C}^{k_{max} \times O \times I}$, $\boldsymbol{U} \in \mathbb{R}^{n \times I}$, $\hat{\mathcal{F}}(\boldsymbol{U}) \in \mathbb{C}^{[\frac{n}{2}]+1 \times I}$ and $\hat{\mathcal{F}}$ denotes the one-dimensional Fast Fourier Transform for real-valued functions (RFFT). We calculate the product $\mathbf{R} \cdot \hat{\mathcal{F}}(\boldsymbol{U})$ in the way we truncate the higher order modes in $\hat{\mathcal{F}}(\boldsymbol{U})$ and obtain $\hat{\mathcal{F}}(\boldsymbol{U}) \in \mathbb{C}^{k_{max} \times I}$ then we can apply the straightforward modification of equation (8), i.e.*

$$\left(\mathbf{R} \cdot \left(\hat{\mathcal{F}}(\boldsymbol{U})\right)\right)_{k,l} = \sum_{j=1}^{I} R_{k,l,j}\left(\hat{\mathcal{F}}(\boldsymbol{U})\right)_{k,j},$$

*where $k = 1, \ldots, k_{max}$, $l = 1, \ldots, O$. The product $\mathbf{R} \cdot \hat{\mathcal{F}}(\boldsymbol{U}) \in \mathbb{C}^{k_{max} \times O}$ must be padded with zeros in order to bring it to the appropriate shape, i.e. $\mathbf{R} \cdot \hat{\mathcal{F}}(\boldsymbol{U}) \in \mathbb{C}^{[\frac{n}{2}]+1 \times O}$ before applying the inverse of the RFFT operator.*

*Remark:* we use the Fourier Neural Operator expression for the Discrete Fourier Neural Operator in the following.

## 2.2 CONVOLUTIONAL FOURIER LAYER

We use the Discrete Fourier Neural Operator in our proposed model in a very similar way to usual convolution layers. The layer corresponding to the operator has three parameters: the number of the input channels $I$, the number of the output channels $O$, and the size of the kernel truncation $k_{\max}$. In the network architecture, we refer to a Fourier layer as the parallel coupling of a layer corresponding to a Discrete Fourier Neural Operator and a conventional one-dimensional convolution layer with the corresponding padding size to get the same output size, see Figure 1. The motivation behind this coupling is the following. Filters in convolutional neural networks are usually local. They are good at capturing local patterns. The filters in the Fourier Neural Operator are global functions, therefore we expect they are good at capturing global patterns.

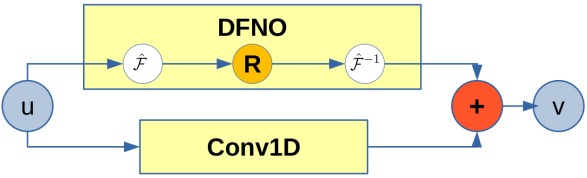

Figure 1: The Convolutional Fourier layer is a parallel coupling of a layer corresponding to the Discrete Fourier Neural Operator and a conventional one-dimensional convolution layer with the corresponding padding size.

## 2.3 PROPOSED *CF-ResNet-9-1D* MODEL

The model we propose is a simple *ResNet*-type model (He et al., 2016), we call it *CF-ResNet-9-1D*. The total number of trainable parameters is 2.6M-8M. Essentially, we have replaced only the one-dimensional convolution layers with Fourier layers in the simplest *ResNet-9* model, thus obtaining a smaller and larger network, see Figure 2. More technical information about the network is coming now with some explanations for the notations. We use Gaussian Error Linear Units (GELU) as activation functions (Hendrycks & Gimpel, 2016). The further abbreviations on Figure 2 are:

- `FL`$(I, O)$: Fourier Layer with $I$ input channels and with $O$ output channels,
- `GELU`: The GELU activation function
- `BN`$(I)$: Batch normalization in $I$ channels,

- $\texttt{FC}(I, O)$: Fully connected layer between $I$ and $O$ neurons.
- $\texttt{avgpool,}\,2$: Performs average pooling operation on the input by the kernel size 2, this is also the size of the stride here.
- $\texttt{AdaptiveAvgPool1d}(1)$: Performs adaptive average pooling operation, in this case, the size of the output is 1 by each channel.
- $\texttt{num\_classes}$: number of classes at the output layer.

Other important parameters of the layers are:

- The size of the truncation in the Fourier Layers are the same with the parameter $k_{\max} = 16$.
- The size of the kernel is 11 and the size of the padding is 5 in the one-dimensional convolution layers.

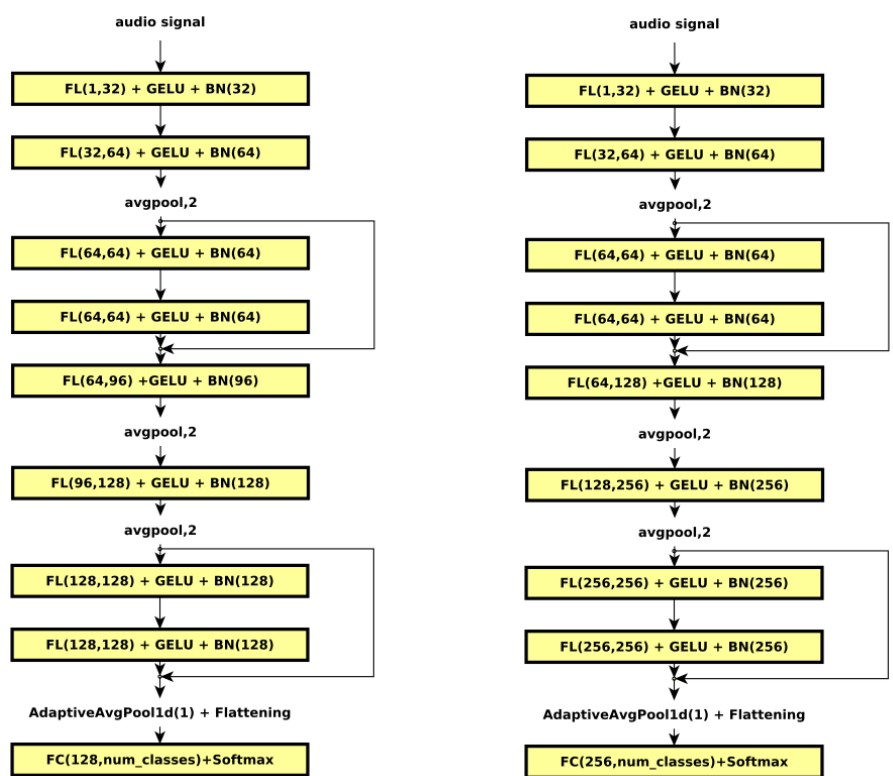

Figure 2: The architecture of the proposed small and large *CF-ResNet-9-1D* models.

Furthermore, the two sideways arrows in the figure represent the residual connections.

## 2.4 DATASETS

This study utilizes four publicly available dataset whose names are *Wingbeats* (Potamitis & Rigakis, 2016), *Fruitflies* (Rigakis et al., 2019), *Insects* (Chen et al., 2014b) and *Abuzz* (Mukundarajan et al., 2017).

The *Wingbeats* and *Abuzz* datasets contain raw audio signals of six mosquito species of three different genera, namely these are *Ae. aegypti*, *Ae. albopictus*, *An. arabiensis*, *An. gambiae*, *Cu. pipiens*, and *Cu. quinquefasciatus*. The data were collected individually at the premises of Biogents (Regensburg, Germany) and recorded by large aperture optoelectronic devices in the *Wingbeats* dataset. Each recorded audio sound is 0.65s long or has a length of 5000 samples at 8KHz sampling rate.

The total number of records is $279556$. The dataset is imbalanced, the details can be seen in Table 5.

The *Fruitflies* dataset was collected from Gouves and Chersonisos areas in Crete using the same collection method as the *Wingbeats* dataset. It comprises audio signals from three distinct species of fruit flies: *Drosophila melanogaster*, *Drosophila suzukii*, and *Zaprionus*. The signals were sampled at 8kHz. The dataset contains $34,518$ recordings, each lasting $0.65$ seconds. The class sizes are shown in Appendix, Table 6.

The *Insects* dataset was generated by the UCR Computational Entomology Group (Chen et al., 2014b) collected in a similar manner using pseudo-acoustic optical sensors. It contains 10 classes, namely *Aedes aegypti (female)*, *Aedes aegypti (male)*, *Drosophila simulans*, *Musca domestica*, *Cx. quinquefasciatus (female)*, *Cx. quinquefasciatus (male)*, *Cx. stigmatosoma (female)*, *Cx. stigmatosoma (male)*, *Cx. tarsalis (female)* and *Cx. tarsalis (male)*. The dataset contains a total of $50,000$ recordings, each $0.1$s long sampled at 6kHz, and each class consists of $5,000$ elements.

In the *Abuzz* dataset the data were collected by mobile phones. Originally, the length of the recordings varies up to 5 min, with sample rates of 8000Hz and $44,100$Hz. Here, we use the pre-processed dataset from the article (Wei et al., 2022). This means that we have 10s long signals. The total amount of recording is $915$, and the element numbers of each class can be seen in Appendix, Table 5. During the numerical experiments, we converted each signal at 8000Hz sampling rate.

## 2.5 DATA PREPROCESSING

In all scenarios, we consistently utilize raw audio signals. We aim to compare our results fairly with those presented by Wei et al. (2022). Following the methodology Fanioudakis et al. (2018), we partition the dataset exactly the same way into training and testing sets.

The learning set contains 80% of the data and it is further divided into training and validation sets in a stratified fashion for cross-validation. This means that we consider a $60/20/20\%$ split for the training/validation/test sets, the exact sizes of the sets can be seen in Table 1. Then, we normalize the data as simply as possible, which means that the entire dataset is considered as the measured data of one variable. Therefore, we perform simple standardization by calculating the two required scalars, the mean and the standard deviation of the training set.

Table 1: Sizes of the training, validation, and testing sets.

| Dataset | Training | Validation | Testing |
|---|---|---|---|
| *Wingbeats* | $167,739$ | $55,914$ | $55,913$ |
| *Abuzz* | $549$ | $184$ | $182$ |
| *Fruitflies* | $20,710$ | $6,904$ | $6,904$ |
| *Insects* | $30,000$ | $10,000$ | $10,000$ |

## 2.6 TRAINING

A stochastic Gradient optimizer is used with Nesterov momentum during the training. We also use the One-cycle learning rate scheduler (Smith & Topin, 2017). The setting and choosing of the learning rate scheduler was also a crucial part of our work.

The further parameter settings for the training can be seen in Table 2. These parameters were chosen by hand among evaluating the model with different settings on the validation set, to avoid overfitting and underfitting. This is also true for the selection of the optimizer. As the evaluation metric on the test set, we use accuracy. More precisely, this means that the model with the best validation accuracies during training was saved and then evaluated on the test set. All models were implemented on *NVIDIA GeForce RTX 3090 GPU* of 24GB of memory, using *Python3.8* with supported libraries of *PyTorch*, *Librosa*, *Pandas*, and *NumPy*.

Table 2: Parameter settings in the training processes.

| Dataset | Batch size | Epochs | Learning rate | Weight decay |
|---------|------------|--------|---------------|--------------|
| *Wingbeats* | 32 | 25 | 0.0005 | 0.005 |
| *Abuzz* | 4 | 150 | 0.0005 | 0.005 |
| *Fruitflies* | 32 | 25 | 0.0002 | 0.005 |
| *Insects* | 32 | 25 | 0.0005 | 0.005 |

## 3 RESULTS

We found that our proposed *CF-ResNet-9-1D* models have outperformed other models on the *Wingbeats*, the *Fruitflies* and the *Insects* datasets, i.e. the *WbNet* (Wei et al., 2022) and even *DenseNet-121*, *MobileNet* models (Fanioudakis et al., 2018). We summarize these results in Table 3, the two largest values are written in bold in each column.

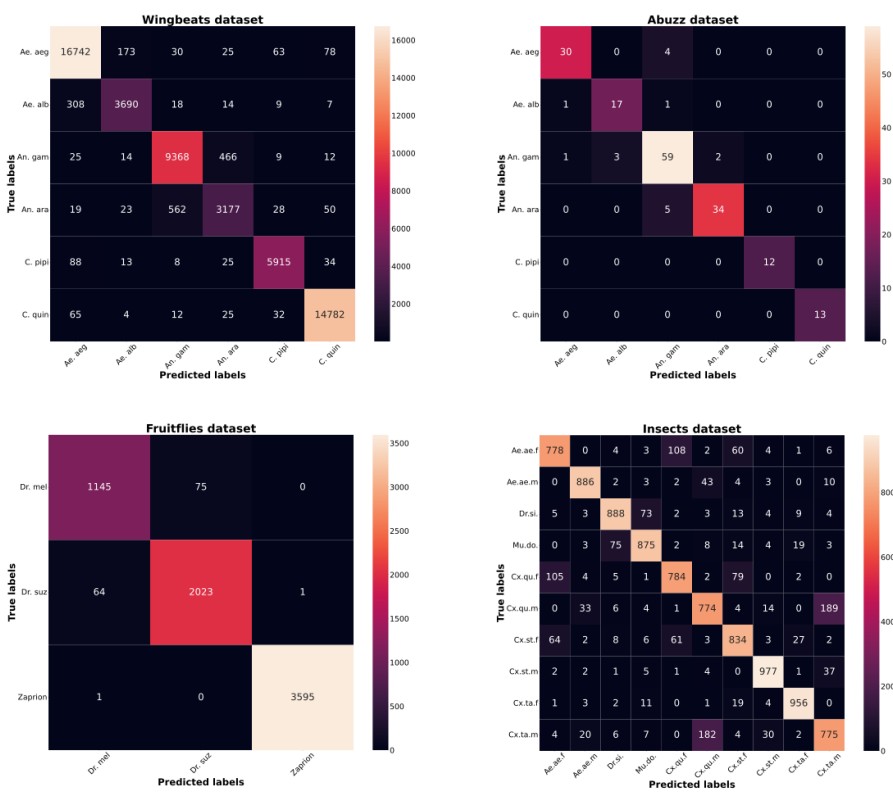

Figure 3: Confusion matrices on the testing datasets for the large *CF-ResNet-9-1D* model which has the best validation accuracy among 5 different runs.

The proposed model was also evaluated without the Fourier layer in the numerical experiments as well, so that to substantiate the utility of the Fourier layer. To be more specific, the numerical experiments were performed using both small (with 670K trainable parameters) and large (with 8.0M parameters) *ResNet-9* models, which include one-dimensional convolutional layers. The specifics of these *ResNet-9* models can be found in the figure 5 included in the Appendix.

The small and large *CF-ResNet-9-1D* models achieved 95.87% and 96.01% average test accuracy on the *Wingbeats* dataset among five different runs. The average test accuracies achieved by the

Table 3: Accuracies for different architectures. The abbreviations stand for the following: *TP* - the number of the trainable parameters, *BVA* - the best validation accuracy over five different runs, *TA* - the corresponding test accuracy, *ATA* - average test accuracy, *RS* - raw samples, *SP* - spectrogram.

| Architecture | TP | Data | BVA % | TA % | ATA % |
|---|---|---|---|---|---|
| *WINGBEATS* | | | | | |
| small *CF-ResNet-9-1D* (ours) | 2.6M | *RS* | **96.08** | **95.92** | **95.87** |
| large *CF-ResNet-9-1D* (ours) | 7.8M | *RS* | **96.12** | **95.99** | **95.98** |
| small vanilla *FNO-ResNet-9* | 2.0M | *RS* | 88.24 | 88.27 | 88.25 |
| large vanilla *FNO-ResNet-9* | 6.0M | *RS* | 88.77 | 88.61 | 88.53 |
| small *ResNet-9* [1] | 0.7M | *RS* | 95.55 | 95.43 | 95.37 |
| large *ResNet-9* [1] | 8M | *RS* | 95.65 | 95.35 | 95.43 |
| *DenseNet121* (Fanioudakis et al., 2018) [2] | 7M | *SP* | 92.16 | 91.97 | 91.92 |
| *MobileNet* (Fanioudakis et al., 2018) [2] | 2M | *SP* | 91.41 | 91.14 | 91.20 |
| *WbNet* (Wei et al., 2022) [2] | 11M | *SP* | 87.62 | 91.09 | 90.32 |
| *ABUZZ* | | | | | |
| small *CF-ResNet-9-1D* (ours) | 2.6M | *RS* | 93.48 | 86.26 | 85.05 |
| large *CF-ResNet-9-1D* (ours) | 7.8M | *RS* | 95.11 | 90.66 | 86.59 |
| small vanilla *FNO-ResNet-9* | 2.0M | *RS* | 63.04 | 57.69 | 52.42 |
| large vanilla *FNO-ResNet-9* | 6.0M | *RS* | 62.50 | 48.35 | 54.39 |
| small *ResNet-9* [1] | 0.7M | *RS* | 95.11 | 87.36 | 85.49 |
| large *ResNet-9* [1] | 8M | *RS* | 97.28 | 92.86 | 90.33 |
| *DenseNet121* Fanioudakis et al. (2018) [2] | 7M | *SP* | **99.79** | **97.09** | **95.94** |
| *MobileNet* Fanioudakis et al. (2018) [2] | 2M | *SP* | **100.00** | 92.98 | 93.69 |
| *WbNet* Wei et al. (2022) [2] | 11M | *SP* | 62.50 | 73.63 | 67.25 |
| *FRUITFLIES* | | | | | |
| small *CF-ResNet-9-1D* (ours) | 2.6M | *RS* | **98.49** | **97.99** | **97.96** |
| large *CF-ResNet-9-1D* (ours) | 7.8M | *RS* | **98.49** | **97.96** | **98.00** |
| small vanilla *FNO-ResNet-9* | 2.0M | *RS* | 93.60 | 93.86 | 93.45 |
| large vanilla *FNO-ResNet-9* | 6.0M | *RS* | 93.63 | 93.61 | 93.53 |
| small *ResNet-9* [1] | 0.7M | *RS* | 98.16 | 97.75 | 97.64 |
| large *ResNet-9* [1] | 8M | *RS* | 98.32 | 97.71 | 97.67 |
| *DenseNet121* Fanioudakis et al. (2018) [2] | 7M | *SP* | 92.65 | 92.68 | 92.91 |
| *MobileNet* Fanioudakis et al. (2018) [2] | 2M | *SP* | 91.73 | 91.01 | 91.57 |
| *WbNet* Wei et al. (2022) [2] | 11M | *SP* | 86.37 | 86.15 | 86.67 |
| *INSECTS* | | | | | |
| small *CF-ResNet-9-1D* (ours) | 2.6M | *RS* | **85.27** | **85.44** | 85.10 |
| large *CF-ResNet-9-1D* (ours) | 7.8M | *RS* | **85.31** | **85.27** | **85.24** |
| small vanilla *FNO-ResNet-9* | 2.0M | *RS* | 75.59 | 74.98 | 74.81 |
| large vanilla *FNO-ResNet-9* | 6.0M | *RS* | 76.00 | 75.62 | 75.28 |
| small *ResNet-9* [1] | 0.7M | *RS* | 84.91 | 84.63 | 84.73 |
| large *ResNet-9* [1] | 8M | *RS* | 85.26 | 85.18 | **85.16** |
| *DenseNet121* Fanioudakis et al. (2018) [2] | 7M | *SP* | 81.77 | 81.59 | 81.36 |
| *MobileNet* Fanioudakis et al. (2018) [2] | 2M | *SP* | 79.25 | 79.28 | 78.79 |
| *WbNet* Wei et al. (2022) [2] | 11M | *SP* | 76.29 | 78.11 | 77.91 |

large *CF-ResNet-9-1D* model were $85.05\%$, $98.00\%$ and $85.24\%$ on the *Abuzz*, *Fruitflies* and on the *Insects* datasets respectively.

---

[1] According to another accepted article by the authors.

[2] Reevaluated experiment solving data leak in the original implementation.

These five results were obtained by saving the model with the best validation accuracy during the training process and then evaluating them on the test set. Importantly, while *CF-ResNet-9-1D* does not achieve the highest average test accuracy on the Abuzz dataset, it outperforms the other models on the remaining two datasets. The evaluation on the validation set was performed on every 1000 training minibatches in each epoch for the *Wingbeats* dataset, while for the *Abuzz*, the *Fruitflies* and the *Insects* datasets, the model was evaluated after every 30, 100 and 200 minibatches, respectively. The evolution of the accuracies on the training and the validation sets is shown in Figures 4.

The exact details about the test results based on each species can be seen in Table 7. These results were obtained from the models with the best validation accuracy. In Figures 3, we present the confusion matrices for the large *CF-ResNet-9-1D* model. It can be observed that the structures are strongly diagonal. The classes are arranged by genus in pairs, i.e. *Anopheles*, *Aedes* and *Culex* for the *Wingbeats* and the *Abuzz* datasets. There are also two species of the same genus in *Fruitflies*, namely *Drosophila*. Most misclassifications occur between species of the same genus. For the Insects dataset, the larger *CF-ResNet-1D* network is not significantly more efficient than other models as it is for Wingbeats and Fruitflies, perhaps the spectral properties of the recordings are not as dominant in this case. Nevertheless, we can see from Table 4 that both training and inference are substantially faster with our proposed *CF-ResNet-1D* models compared to, for example, the second-best performing *ResNet* model. We can also observe from Table 7 that the CF-ResNet-1D network tends to overly predict two classes in both cases, specifically the *Cx. quinq. (male)* and *Cx. tarsalis (male)* classes at the expense of accuracy. In the case of the *Insects* dataset, most of the confusion occurred between the *Cx. quinquefasciatus (male)* and *Cx. tarsalis (male)* classes. Further complicating the classification here is the presence of samples from the same species that included both female and male individuals in the dataset.

Table 3 also illustrates that the conventional convolution with a kernel size of 1 in the original approach proposed by Li et al. (2021) is insufficient for time series processing. We refer to this network as vanilla *FNO-ResNet-9* here, in order to clearly distinguish it from the *CF-ResNet-9-1D* architecture we propose. As the results indicate Convolutional Fourier Layers improve significantly on traditional FNO layers.

In general, our proposed model performed better for the *Wingbeats*, *Fruitflies* and *Insects* datasets compared to the *Abuzz* dataset. Of course, this may also be due to the fact that these datasets are significantly larger than the *Abuzz*. It is also important to note that the *Wingbeats*, *Fruitflies* and *Insects* audio recordings were short, only 0.65s and 0.1s long, and captured using advanced audio equipment, while the *Abuzz* data were longer and recorded using mobile devices in noisy environments, varying up to 5 minutes in length. Therefore, *Abuzz* audio recordings were divided into multiple 10s long segments.

## 4 CONCLUSION

In this study, we presented a novel approach for incorporating the Fourier Neural Operator in Deep Learning models for time series classification, while also providing a definition of the operator more fitting to this task.

We proposed *CF-ResNet-9-1D* a ResNet-like model that consists of Convolutional Fourier (CF) Layers that highly improve on the original Fourier Neural Operators by adding a parallel local convolution. Experimental evaluation is carried out on insect wingbeat sound classification datasets where our models achieve a new state-of-the-art result on the *Wingbeats*, *Fruitflies*, and *Insects* datasets, outperforming previously published models, while also maintaining a competitive inference and training speed. To aid mosquito-borne disease control and intervention efforts we release our implementation and pre-trained models.

As combining FNOs with wide-spread deep learning operators empirically yielded improvements over current state-of-the-art, exploring the effects of data augmentation, and more complex Fourier domain processing, such as adaptive filtering, attention, or Short-Time Fourier Transform-based solutions are promising questions for further research.

## 5 Reproducibility

We set all possible random seeds during our numerical experiments. To account for the variability of random processes we repeat each experiment 5 times and report the best and average scores in Section 3. To retain anonymity the supporting code for our experiments and the open-source datasets are made available as supplementary material only. The camera-ready version will include a public repository with the same material.

## 6 Ethics Statement

This research does not include any potentially harmful datasets, methods, or bias, nor any human-related experiments. We declare that the authors have no conflicting interests or funding, that could influence the research procedure or results.

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

# A  APPENDIX

Table 4: Sample processing speeds of different architectures using the largest power of two batch sizes which fit into $24GB$ memory. *TP* denotes the number of the trainable parameters, *BS* marks the batch size, *TTPS* is training time per sample in milliseconds, *ITPS* denotes inference time per sample in milliseconds, *RS* means raw samples and *SP* shortens spectogram. Processing speeds were averaged over a full epoch of training and validation.

| WINGBEATS | | | | | |
|---|---|---|---|---|---|
| Architecture | TP | Features | BS | TTPS | ITPS |
| small *CF-ResNet-9-1D* (ours) | 2.6M | RS | 512 | 0.72 | 0.21 |
| large *CF-ResNet-9-1D* (ours) | 7.8M | RS | 512 | 0.92 | 0.29 |
| small *ResNet-9* [3] | 0.7M | RS | 512 | **0.45** | **0.14** |
| large *ResNet-9* [3] | 8M | RS | 256 | 1.83 | 0.43 |
| *DenseNet121* Fanioudakis et al. (2018) [4] | 7M | SP | 256 | 0.85 | 0.30 |
| *MobileNet* Fanioudakis et al. (2018) [4] | 2M | SP | 512 | **0.38** | **0.14** |
| *WbNet* Wei et al. (2022) [4] | 11M | SP | 128 | 1.20 | 0.38 |
| ABUZZ | | | | | |
| Architecture | TP | Features | BS | TTPS | ITPS |
| small *CF-ResNet-9-1D* (ours) | 2.6M | RS | 64 | 13.42 | 3.84 |
| large *CF-ResNet-9-1D* (ours) | 7.8M | RS | 32 | 17.78 | 5.53 |
| small *ResNet-9* [3] | 0.7M | RS | 64 | **3.00** | **1.08** |
| large *ResNet-9* [3] | 8M | RS | 32 | 20.97 | 5.11 |
| *DenseNet121* Fanioudakis et al. (2018) [4] | 7M | SP | 32 | 13.08 | 4.20 |
| *MobileNet* Fanioudakis et al. (2018) [4] | 2M | SP | 32 | **5.40** | **1.86** |
| *WbNet* Wei et al. (2022) [4] | 11M | SP | 8 | 22.04 | 5.11 |
| FRUITFLIES | | | | | |
| Architecture | TP | Features | BS | TTPS | ITPS |
| small *CF-ResNet-9-1D* (ours) | 0.7M | RS | 512 | 0.69 | 0.21 |
| large *CF-ResNet-9-1D* (ours) | 8M | RS | 512 | 0.96 | 0.28 |
| small *ResNet-9* [3] | 0.7M | RS | 512 | **0.44** | **0.14** |
| large *ResNet-9* [3] | 8M | RS | 256 | 1.92 | 0.40 |
| *DenseNet121* Fanioudakis et al. (2018) [4] | 7M | SP | 256 | 1.67 | 0.56 |
| *MobileNet* Fanioudakis et al. (2018) [4] | 2M | SP | 512 | **0.37** | **0.13** |
| *WbNet* Wei et al. (2022) [4] | 11M | SP | 128 | 1.22 | 0.42 |
| INSECTS | | | | | |
| Architecture | TP | Features | BS | TTPS | ITPS |
| small *CF-ResNet-9-1D* (ours) | 2.6M | RS | 8192 | 0.13 | **0.03** |
| large *CF-ResNet-9-1D* (ours) | 7.8M | RS | 4096 | 0.16 | 0.04 |
| small *ResNet-9* [3] | 0.7M | RS | 8192 | **0.08** | 0.04 |
| large *ResNet-9* [3] | 8M | RS | 2048 | 0.24 | 0.07 |
| *DenseNet121* Fanioudakis et al. (2018) [4] | 7M | SP | 512 | 0.57 | 0.18 |
| *MobileNet* Fanioudakis et al. (2018) [4] | 2M | SP | 4096 | **0.09** | **0.03** |
| *WbNet* Wei et al. (2022) [4] | 11M | SP | 128 | 13.24 | 0.46 |

---

[3]According to another accepted article by the authors.
[4]Reevaluated experiment solving data leak in the original implementation.

Table 5: Element numbers of each class in the *Wingbeats* and in the *Fruitflies* datasets.

| Species | Wingbeats | Abuzz |
|---|---|---|
| *Ae. aegypti* | 85553 | 324 |
| *Ae. Albopictus* | 20231 | 197 |
| *An. Gambiae* | 49471 | 171 |
| *An. Arabiensis* | 19297 | 95 |
| *Cu. pipiens* | 30415 | 66 |
| *Cu. quinquefasciatus* | 74599 | 62 |

Table 6: Element numbers of each class in the *Fruitflies* dataset.

| Species | Fruitflies |
|---|---|
| *Dr. melanogaster* | $6,064$ |
| *Dr. suzukii* | $10,142$ |
| *Zaprionus* | $18,312$ |

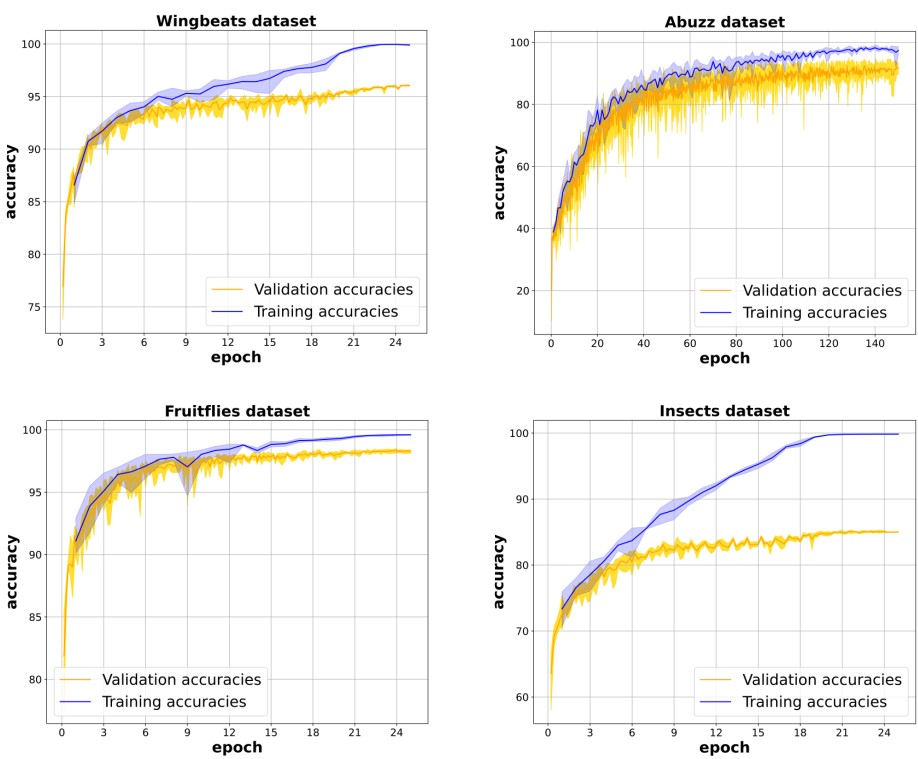

Figure 4: Performance on the different datasets for the accuracy generated from 5 independent runs by the large *CF-ResNet-9-1D* model. The shaded region is enclosed between the maximum and minimum values over the runs, while the boldface curve displays the average.

Table 7: The *CF-ResNet-9-1D* models evaluation metrics on the different datasets.

| WINGBEATS | | | | | | |
|---|---|---|---|---|---|---|
| | Small *CF-ResNet-9-1D* | | | Large *CF-ResNet-9-1D* | | |
| Species | Precision | Recall | F1-score | Precision | Recall | F1-score |
| *Ae. aegypti* | 97.07 | 97.83 | 97.45 | 97.07 | 97.84 | 97.45 |
| *Ae. Albopictus* | 94.28 | 91.32 | 92.78 | 94.20 | 91.20 | 92.68 |
| *An. Gambiae* | 93.18 | 94.55 | 93.91 | 93.70 | 94.68 | 94.19 |
| *An. Arabiensis* | 84.41 | 81.24 | 82.79 | 85.13 | 82.33 | 83.70 |
| *Cu. pipiens* | 97.67 | 97.25 | 97.46 | 97.66 | 97.22 | 97.44 |
| *Cu. quinq.* | 98.92 | 99.12 | 99.02 | 98.78 | 99.07 | 98.93 |
| ABUZZ | | | | | | |
| | Small *CF-ResNet-9-1D* | | | Large *CF-ResNet-9-1D* | | |
| Species | Precision | Recall | F1-score | Precision | Recall | F1-score |
| *Ae. aegypti* | 88.24 | 88.24 | 88.24 | 93.75 | 88.24 | 90.91 |
| *Ae. Albopictus* | 77.78 | 73.68 | 75.68 | 85.00 | 89.47 | 87.18 |
| *An. Gambiae* | 81.16 | 86.15 | 83.58 | 85.51 | 90.77 | 88.06 |
| *An. Arabiensis* | 88.89 | 82.05 | 85.33 | 94.44 | 87.18 | 90.67 |
| *Cu. pipiens* | 100 | 100 | 100 | 100 | 100 | 100 |
| *Cu. quinq.* | 100 | 100 | 100 | 100 | 100 | 100 |
| FRUITFLIES | | | | | | |
| | Small *CF-ResNet-9-1D* | | | Large *CF-ResNet-9-1D* | | |
| Species | Precision | Recall | F1-score | Precision | Recall | F1-score |
| *Dr. melanog.* | 94.20 | 94.59 | 94.40 | 94.63 | 93.85 | 94.24 |
| *Dr. suzukii* | 96.83 | 96.60 | 96.72 | 96.43 | 96.89 | 96.66 |
| *Zaprionus* | 99.94 | 99.94 | 99.94 | 99.97 | 99.97 | 99.97 |
| INSECTS | | | | | | |
| | Small *CF-ResNet-9-1D* | | | Large *CF-ResNet-9-1D* | | |
| Species | Precision | Recall | F1-score | Precision | Recall | F1-score |
| *Ae. aegypti (female)* | 81.53 | 81.78 | 81.65 | 81.13 | 80.54 | 80.83 |
| *Ae. aegypti (male)* | 92.09 | 92.86 | 92.48 | 92.68 | 92.97 | 92.82 |
| *Drosophila simulans* | 86.61 | 87.65 | 87.13 | 89.07 | 88.45 | 88.76 |
| *Musca domestica* | 86.59 | 87.54 | 87.06 | 88.56 | 87.24 | 87.90 |
| *Cx. quinq. (female)* | 83.72 | 78.00 | 80.76 | 81.58 | 79.84 | 80.70 |
| *Cx. quinq. (male)* | 76.84 | 78.34 | 77.58 | 75.73 | 75.51 | 75.62 |
| *Cx. stigma. (female)* | 80.38 | 84.36 | 82.32 | 80.89 | 82.57 | 81.72 |
| *Cx. stigma. (male)* | 94.90 | 93.98 | 94.44 | 93.67 | 94.85 | 94.26 |
| *Cx. tarsalis (female)* | 96.21 | 94.18 | 95.19 | 94.00 | 95.89 | 94.94 |
| *Cx. tarsalis (male)* | 76.46 | 76.02 | 76.24 | 75.54 | 75.24 | 75.39 |

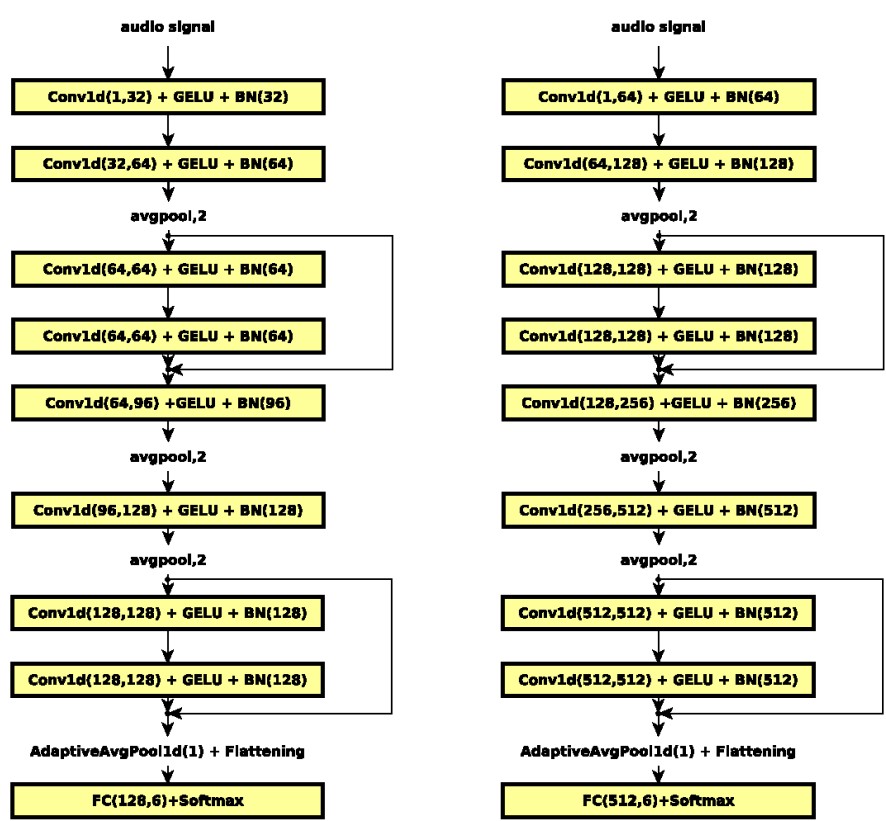

Figure 5: The architecture of the small and large *ResNet-9* models.

