# OpenReview forum: "From PDEs to Wingbeats: A Novel Convolutional Fourier Layer-based ResNet Model"
_ICLR.cc/2024/Conference — ICLR 2024 Conference Withdrawn Submission_

### Official Review · Reviewer_QZGG · 2023-10-24

**Soundness:** 3 good
**Presentation:** 3 good
**Contribution:** 2 fair
**Rating:** 5
**Confidence:** 4

**Summary:**

The authors considers Fourier layers in audio based insect classification problems. Their main contribution seems to be applying both Fourier and conventional convolutional layers in parallel fashion, plus the new application for Fourier layers.

**Strengths:**

+ Application they consider is well presented and nicely motivated
+ Results demonstrates performance of their approach even when it's the proposed method is not that good

**Weaknesses:**

- Methodological contribution is quite thin: concatenating conventional convolutional layer with DFNO (Figure 1). They also claim that they give a new definition for Fourier layer, but it mostly seems to be a different way to describe the original Fourier layers in Li et al 2021 (please correct me if I am wrong). On the other hand, the new application for Fourier layer adds up to the contribution. They also fix data leakage issues in the previous studies which adds up to total contribution (overall this is NOT completely "Weakness" but I do not know how to write this point with this very limiting review submission form)

**Questions:**

- Definition 1: to be more precise, you could say that $\hat F$ denotes one dimensional discrete Fourier transformation. The Fast Fourier transformation is an algorithm which implements the one dimensional discrete Fourier transformation in efficient manner. As you also mention padding and truncation for FFT/IFFT, you could also be more explicit with this using conjugate symmetry i.e. why there is [n/2]+1 instead of n.
- Figure 2 could somehow include "n" or "resolution" dimension. This would make effect of pooling layers more visible and also to see how resolution is decreasing.
- Section 2.3: "FL(I,O): Fourier layer with ..." you could add "convolution" to make more clear that this actually includes the parallel convolution

- In results, the improvement of their method compared to traditional ResNets is not that significant. It could be discussed that when the improvement is sufficient considering added computational complexity. For example, is it worthwhile to utilize Fourier layers instead of normal ResNets in mobile applications?

---

### Official Review · Reviewer_EZRu · 2023-10-30

**Soundness:** 3 good
**Presentation:** 2 fair
**Contribution:** 1 poor
**Rating:** 3
**Confidence:** 5

**Summary:**

This paper added a downstream regressor to the Fourier Neural Operator (FNO), and then applied the model to a classification problem. Given the current form of the paper, it is far from the ICLR caliber as the methodology itself (neural architecture, how to process audio data) adds nothing new to the table.

**Strengths:**

- The paper studies an important problem that seems requiring quite a few engineering tricks to get good data. I am not a biologist or engineer who works on the techniques to get the data, but I went to read Mukundarajan et al. 2017 (where the authors got the Abuzz dataset), and am quite amazed by the endeavor therein.
- This paper opts for the standard train-valid-test split to correct a data leakage in Wei et al. 2022 paper (I am assuming by the writings, therein the optimizers' early stop criterion is based on the performance on the test set, then the result is reported on the test set).

**Weaknesses:**

- In section 2.1, the authors rewrote the matrix-valued kernel integral's convolution theorem in the FNO paper to a discrete summation form explicitly. I am not sure how this is anything new, as even the convolution theorem's DFT version is normally an exercise in a graduate-level DSP class, e.g., see Mallat's book. Given a non-idiomatic opening statement of this section, "in this motivation part, rigorous mathematical precision is ignored", then why bothering presenting such a subsection?
  - After restricting on $(-\pi, \pi)$, periodicity has to be assumed.
  - Most likely being a copy-paste artifact, the functions of interest are firstly assumed in $L^1$ yet later in $L^2$ on page 3. The convolution operation can be minimally defined for functions in $L^1$, but does not mean in the context of presenting FNO that one should use such a space.
  - After presenting the vectorial setup, equation (3) switches back to scalar-valued functions.
  - In fact, $u$'s Fourier modes are not truncated (this is important for the universal approximator application in the spatial domain), and neither is $u$ parametrized (layerwise speaking).
- With the point above being said, I think section 2.1 is a very good practice for the student author to present in his/her thesis. However, it is rather pointless to feature it in an ICLR submission without any architectural change or new interpretation of the FNO architecture.
- In fact, the original FNO ***IS*** implemented exactly as what is shown in Figure 1. `Conv1d` is used in the original code of FNO, it is just that when the Caltech group wrote the paper, they used a letter "W" for this mechanics while including the possibility of a non-pointwise universal approximator. For example, please check the `fno_1d.py` in the `master` branch of the FNO repo (https://github.com/neuraloperator/neuraloperator/blob/master/fourier_1d.py).
- The authors said "the motivation behind this coupling is the following: ... the filters in the Fourier Neural Operator are global functions, therefore we expect they are good at capturing global patterns." This is completely wrong. The `Conv1d` in the original FNO code is **pointwise** (kernel size being 1), exactly the same as the MLP in Transformer (e.g., see Guibas et al. ICLR 2022 paper and their mechanics in FNOs/Transformers are explained/explored in the references cited therein).
- The proposed CF-ResNet model is using the template of feature extractor + tensor2tensor + downstream regressor (flattening). There is nothing new in either component. The first layers for the vanilla FNO-based models with raw time series input simply expands the number of channels using linear transformation with random initialization. Of course, the temporal correlation nature of the data makes the size 1 kernel FNO underperform versus the ones with larger-than-1 kernel size `Conv1d`. This comparison is rather unfair for the vanilla FNO-based models, without the standard procedure to add more channels using standard filter banks.
- This is minor, but mostly likely, there is a special character in the bib that triggers the braces not enclosed, which results in the first page's hyperlinks being messed up.

**Questions:**

- There are now a huge repository of neural network approaches for time series data (using the time domain raw input). Yet, why are there no baselines in this regard? Notable examples include WaveNet (and its normalizing flow variant) and Neural ODE (and its variants such as controlled Neural ODE, and latent Neural ODE).
- Why, at least heuristically, do the FNO-based models with a conv kernel size bigger than 1 outperform ***consistently*** the one using the mel spectrogram (I assume that the 2D models use this, and this is what `librosa` is for) on these datasets? The reason of this question is that the spectrograms already encode the information of multiple bands in frequency domain.

---

### Official Review · Reviewer_YK2j · 2023-10-31

**Soundness:** 1 poor
**Presentation:** 2 fair
**Contribution:** 1 poor
**Rating:** 3
**Confidence:** 2

**Summary:**

This work introduces CF-ResNet-1D, a ResNet-inspired model which is built from Convolutional Fourier Layers and is consists of parallel units of FNO and 1D-Convolution. Then the authors apply the proposed model for time-series data analysis.

**Strengths:**

To use FNO and CNN for the analysis of time series data is interesting.

**Weaknesses:**

1. The writing of this paper is crude.
2. The authors argue that the motivation of the CF-ResNet-1D is that filters in CNN are local operators and filters in FNO are global operators therefore, the current work try to combine these two operators. While the motivation is not convincing, and the ideal of combining FNO and 1D Convolution is not really novel.
3. The authors should provide theoretical analysis or detailed discussions on the superiority of the proposed model over the conventional one.

**Questions:**

Please refer to the weaknesses part.

---

### Official Review · Reviewer_91ro · 2023-11-01

**Soundness:** 2 fair
**Presentation:** 2 fair
**Contribution:** 2 fair
**Rating:** 3
**Confidence:** 4

**Summary:**

The paper studies the application of Fourier Neural Operators(FNOs), typically applied to solving PDES, in the specific tasks of insect wingbeat sound classification. They propose a Resnet architecture with the novel addition of stacking DFNO layers in parallel with typical 1D convolutional layers. They perform an ablation study on the wingbeat classfication task and compare with other approaches.

**Strengths:**

The paper presents a novel idea: combining FNO layers with Conv 1D layers in the context of a classification task and provides robust evaluation.

**Weaknesses:**

- Table 3 shows that a vanilla ResNet with a similar number of parameters performs equally or better on the average test accuracy than the proposed hybrid architecture. While negative results are worth publishing, the paper does not acknowledge this shortcoming. Based on the results in table 3, the conclusion appears to be that traditional Convolutional ResNet outperform FNO and FNO hybrids, with reduced complexity and computational cost.
- Table 3 lacks confidence intervals. Values that are in the same ballpark should be bolded together, such as the resnet-9 results on Wingbeats, Fruitflies, as was done in Insects (here, the small CF-Resnet should also be bolded).

- While the number of parameters is acknowledged in the table, the number of FLOPS and time to converge should be added to the comparison.
- If the goal of the paper is to convince the community on the benefits of the hybrid architecture, a broader benchmark should be included, with well-studied benchmark audio classification tasks such as the librespeech phone classification task.

**Questions:**

- What motivates selecting the best performing model out of 5 runs? Does it not make more sense to report the confidence interval of the test accuracy out of the 5 runs? Validation accuracy can then be used to perform early stopping if needed.